# Heat conduction measurements in ballistic 1D phonon waveguides indicate breakdown of the thermal conductance quantization

Adib Tavakoli[1,2], Kunal Lulla[1,2], Thierry Crozes[1,2], Natalio Mingo[3], Eddy Collin[1,2] & Olivier Bourgeois[1,2]

Emerging quantum technologies require mastering thermal management, especially at the nanoscale. It is now accepted that thermal metamaterial-based phonon manipulation is possible, especially at sub-kelvin temperatures. In these extreme limits of low temperatures and dimensions, heat conduction enters a quantum regime where phonon exchange obeys the Landauer formalism. Phonon transport is then governed by the transmission coefficients between the ballistic conductor and the thermal reservoirs. Here we report on ultra-sensitive thermal experiments made on ballistic 1D phonon conductors using a micro-platform suspended sensor. Our thermal conductance measurements attain a power sensitivity of 15 attoWatts $\sqrt{\mathrm{Hz}}^{-1}$ around 100 mK. Ballistic thermal transport is dominated by non-ideal transmission coefficients and not by the quantized thermal conductance of the nanowire itself. This limitation of heat transport in the quantum regime may have a significant impact on modern thermal management and thermal circuit design.

[1] Institut NÉEL, CNRS, 25 avenue des Martyrs, 38042 Grenoble, France. [2] Inst NEEL, Univ. Grenoble Alpes, 38042 Grenoble, France. [3] LITEN, CEA-Grenoble, 17 avenue des Martyrs, 38054 Grenoble Cedex 9, France. Correspondence and requests for materials should be addressed to O.B. (email: olivier.bourgeois@neel.cnrs.fr)

**M**anipulating phonons or heat requires specific experimental treatments and theoretical models, especially for low dimensional systems at extremely low temperatures[1–5]. In a one dimensional (1D) quantum channel connecting two reservoirs, the heat current is related to the probability for a phonon of being transmitted from one heat bath to the other when they are kept at different temperatures. This is well described by the Landauer formalism, which expresses thermal conductance in terms of transmission between reservoirs[6–14]. Although many experiments have been done for electrons[15–17] or photons[18,19], very few experimental attempts have been made to probe this limit for phonons[20–22]. This leaves open crucial questions like: where is the temperature drop located, where does the thermal resistance appear, or where do the phonons scatter? This should be clearly expressed through the transmission coefficient that quantifies the connection between the 1D conductor and the reservoir, the mismatch of phonon eigenmodes, and the potential thermal resistance at the contact.

The Landauer expression for the heat flux in a ballistic 1D nanowire between two reservoirs is given by[8–10,23]:

$$\dot{Q} = \sum_{\alpha} \int_0^{\infty} \frac{dk}{2\pi} \hbar \omega_{\alpha}(k) \nu_{\alpha}(k) \left[ \eta_{\mathrm{h}} - \eta_{\mathrm{c}} \right] \mathcal{T}_{\alpha}(k), \qquad (1)$$

where $\alpha$ is the mode index, $\omega_{\alpha}(k)$ is the dispersion relation for the phonon with wave vector $k$, $\nu_{\alpha}(k)$ is the group velocity, $\eta_i(\omega) = 1/(e^{\hbar \omega / k_B T} - 1)$ represents the thermal distribution of phonons in the reservoirs, $k_B$ the Boltzmann constant, and $\mathcal{T}_{\alpha}(k)$ is the transmission coefficient between the two reservoirs.

At low enough temperature, when only fundamental acoustic modes are occupied in the 1D quantum channel connecting the two reservoirs, the heat flux given by Eq. (1) can be expressed through the following approximation:

$$\dot{Q} = N \mathcal{T} G_{\mathrm{q}} (T_2 - T_1) \qquad (2)$$

valid only when $T_2 - T_1 \ll \frac{T_2 + T_1}{2}$, and where $N$ is the number of conducting channels, $\mathcal{T}$ is the transmission coefficient integrated over all the wave-vector, $G_{\mathrm{q}} = \pi^2 k_B^2 T / 3h$ is the universal upper limit of phonon heat transport known as the quantum of thermal conductance for one channel; $T_1$ and $T_2$ being the reservoir's temperature[6–10,20].

Accessing this quantum regime experimentally is highly challenging since ballistic transport of phonons and an optimal coupling of the quantum channels to the two reservoirs are required. This can be obtained at very low temperatures, where the phonon mean free paths (MFP) can be longer than the length of the heat conductor, and the dominant phonon wavelength is much bigger than the nanowire diameter. In this extreme limit, we expect phonons to travel ballistically between the two reservoirs through the 1D waveguide; thermal transport is then dominated by the wave nature of phonons and thus by the transmission coefficient $\mathcal{T}$. To the best of our knowledge, this has never been addressed in previous experiments with as much control on all the thermal flows and characteristic lengths involved as we present here.

Here we report on a highly sensitive experimental study of thermal transport between two large reservoirs through a 1D nanowire. The experimental platform consists of two membranes (thermal reservoirs). The nanowires are micro and nano-engineered from stoichiometric silicon nitride (SiN), the most widely used material in low temperature thermal and mechanical measurements, which behaves like an ideal elastic continuum medium[2,8,24]. The dominant phonon wavelength given by:

$$\lambda_{\mathrm{dom}} = \frac{h \nu_s}{2.82 k_B T} \qquad (3)$$

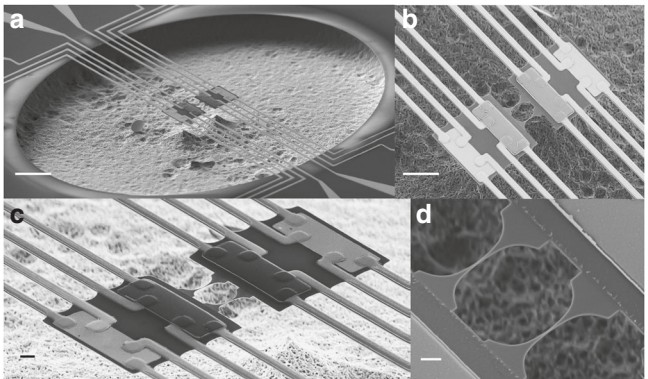

**Fig. 1** Suspended silicon nitride measurement platform. **a** Scanning electron microscopy (SEM) images of the fully suspended silicon nitride membrane-based nanocalorimeters designed for very low temperature measurements. Scale bar, 20 μm. **b**, **c** Ensemble view of the nanocalorimeters with, for each membrane, a copper heater and a NbN thermometer. The thermometers have been purposely installed to probe the temperature very close to the 1D nanowires. A four-contact geometry on the heaters and thermometers allows for highly sensitive electrical measurements of the thermal properties. The two membranes are thermally interconnected via the 1D nanowires. Scale bars are 10 μm in **b** and 2 μm in **c**. **d** Top view of the nanowires obtained by e-beam lithography. The connection of the 1D nanowires to the reservoirs has a catenoidal shape to optimize the value of the transmission coefficient[9]. Scale bar, 1 μm

reaches 200 nm at 1 K in SiN and it still increases as the temperature decreases up to 2 μm at 0.1 K[25–27]; $h$ and $\nu_s$, are, respectively, the Planck constant and the phonon group velocity. We can then assume that a nanowire of diameter $d$ below 100 nm will enter the 1D regime below 1 K (when $\lambda_{\mathrm{dom}} \gg d$). The second characteristic length for heat transport is the phonon mean free path $\Lambda_{\mathrm{ph}}$, which has to be longer than the nanowire length in order to have ballistic transport. This ballistic limit will be probed experimentally by measuring nanowires of different lengths.

## Results

**Experimental set-up for thermal conductance measurements.** The thermal conductance and hence the transmission coefficient are measured as a function of temperature in the ballistic regime. This is done on 1D heat conductors suspended between two 2D membrane reservoirs both controlled in temperature. Importantly, the temperature gradient can be reversed to measure the thermal flow in both directions.

As it is shown in Fig. 1, the measurement platform consists of two adjacent membranes suspended by eight supporting beams and thermally connected to each other by one or two nanowires (the 1D phonon waveguides). Each 1D nanowire has four quantum channels of heat conduction, corresponding to the four different acoustic phonon polarizations (one longitudinal, two transverse and one shear modes)[9]. To reduce any kind of thermal contact resistance due to acoustic mismatch between the nanowire and the membrane, the whole system has been microfabricated with the same material (SiN), with a thickness of 100 nm. The thermal link between the 1D phonon waveguide and the reservoirs is patterned with a catenoidal shape to optimize the transmission of the phonon modes[9]. Based on numerical calculations, this geometrical design was found to maximize the transmission coefficient[9,13]. Each membrane-based calorimeter contains a copper heater and a niobium nitride (NbN) resistive thermometer[28]. For this work, this kind of sensitive thermometry has been adapted to get the best performances at very low temperature, from 30 mK to 10 K.

The heater resistance is precisely measured allowing us to accurately estimate the dissipated power. Thanks to a low noise measurement technique, the double membrane nanocalorimeter has a state of the art power sensitivity of 15 attoWatt $\sqrt{Hz}^{-1}$ at 0.1 K, allowing us to measure the thermal conductance variation with a sensitivity of $1.5 \times 10^{-16}$ W K$^{-1}$ $\sqrt{Hz}^{-1}$ (0.15 femtoWatt K$^{-1}$ $\sqrt{Hz}^{-1}$)[29]. This measurement configuration based on a continuous heat flow between two reservoirs has been used in the past to probe the thermal properties of suspended nanostructures at room temperature[30–33]. This configuration is particularly well suited for thermal measurements in the ballistic limit at very low temperature, when the local temperature cannot be defined along the nanowire; temperature can only be set on large thermodynamic reservoirs like the membranes. There are two major points of difference in this experiment: first, there is no material deposited on top of the suspended nanowire (no parasitic thermal path) and second, the temperature gradient can be reversed between the two membrane sensors to probe the symmetry of the heat flux.

**Measurement protocols and thermal modelling**. The protocols of the different thermal experiments are detailed in Fig. 2. In the first protocol (Fig. 2a), a continuous power is dissipated in the heater of membrane 1. A temperature gradient is then established between the two membranes, the heat flowing through the two nanowires (the two heat waveguides). The direction of the heat flow is reversed in the second protocol where the membrane 2 has a higher temperature than membrane 1 (Fig. 2b), the power being dissipated in the heater of membrane 2. These protocols are essential to probe the symmetry of the heat transport in order to proof the reliability of the measurements.

Figure 2c shows the same steps of Fig. 2a and b after having cut one of the nanowires by focused ion beam (FIB). This decreases the total number of conducting channels from potentially eight quantum channels to only four in the case of one 1D nanowire. Finally, Fig. 2d shows the last protocol enabling the measurement of the thermal conductance of the suspending beams without any nanowire connecting the two membranes. The temperature $T$ corresponds to the average temperature at the two extremities of the nanowires $T = \frac{T_1 + T_2}{2}$, with the condition $|T_1 - T_2| \ll T$ enforced in all the experiments (see Methods). The thermal conductance of the nanowire is obtained using the equation:

$$K_{NW} = K_{b2}\left(\frac{T_2 - T_0}{T_1 - T_2}\right) \qquad (4)$$

where $T_0$ is the temperature of the bath, $K_{b2}$ is the thermal conductance of the suspending beams of each membrane at the temperature $T_2$, $T_1 - T_2$ is of the order of 20 mK and $T_2 - T_0$ of 10 mK. The details of the power balance, calibration, and sensitivity are presented in the Methods section.

**The thermal conductance in the 1D ballistic limit**. The thermal conductance measurements versus temperature along with the data interpretation are presented in Fig. 3 for one and two nanowires. The variation of thermal conductance covers more than three decades as the temperature is lowered down to 50 mK. The superposition of the two measurements (red and black dots) that have been acquired using different directions of the heat flow clearly demonstrates the symmetry of the heat transport. If we now compare the thermal conductance of one and two nanowires, it appears that at high temperature the heat flow for two nanowires is indeed two times the heat flow in one nanowire. As

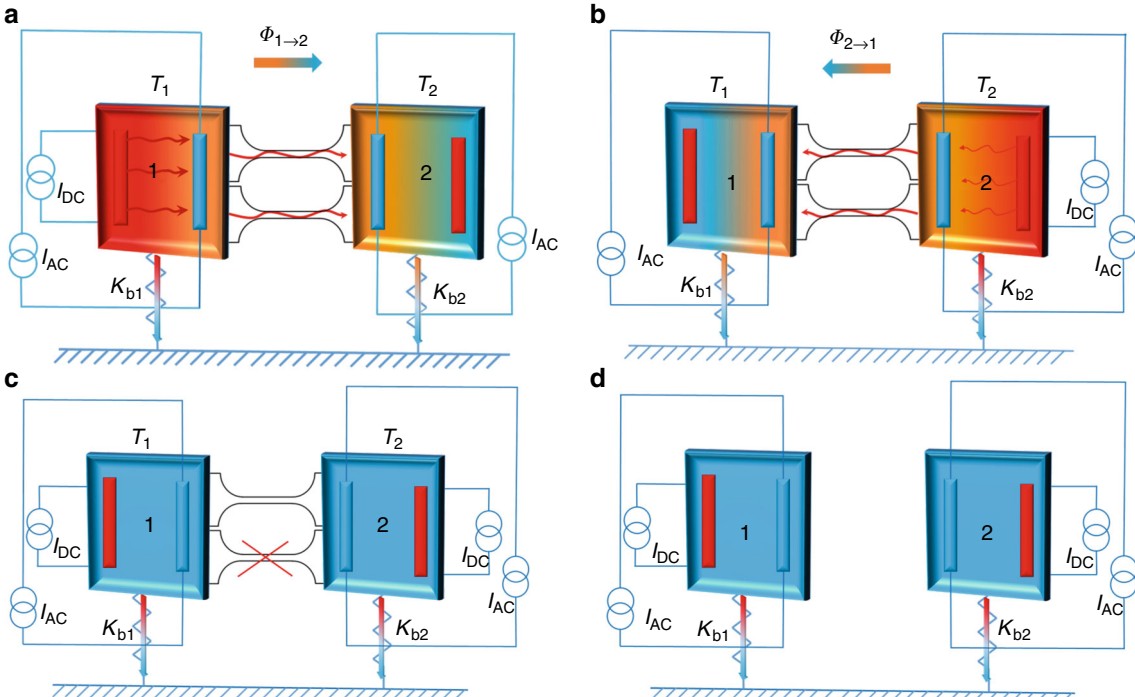

**Fig. 2** Sample design and experimental protocols. The membranes are schematized by a colored square containing a heater in red and a thermometer in blue. Several experimental protocols have been used, **a** where the temperature gradient is established through a DC current applied to the heater on the right membrane creating a continuous heat flow to the left membrane through the 1D phonon waveguide. The two temperatures $T_1$ and $T_2$ are measured with the two thermometers and **b** the DC current is supplied to the heater on the left membrane, the heat flow is then reversed as compared to the case **a**. The two first protocols (**a**, **b**) have been established to verify the symmetry of heat transport in the nanowires. In protocol **c**, one of the nanowires has been cut by FIB, then the steps **a**, **b** are repeated to check the additivity of the heat flow and potentially to probe the number of conducting channels in each nanowire. In **d** both nanowires have been cut to measure the thermal conductance $K_{b1,2}$ of the suspending beams of a single membrane

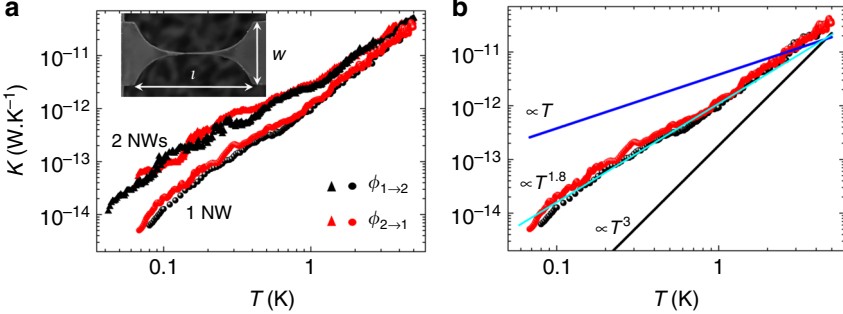

**Fig. 3** Thermal measurements of nanowires as a function of temperature. **a** Thermal conductances of two nanowires (2 NWs) and one nanowire (1 NW) versus temperature. These have been measured in two different heating configurations; red symbols refer to the heating from right to left, and black symbols from left to right. The inset shows the top view of the nanowire illustrating the various dimensions of the nanowire and especially the full length $l$ = 5 μm and the constriction size $w$ = 2.7 μm. **b** The thermal conductance of one nanowire is compared to the theoretical fits. The black line corresponds to the boundary limited regime in $T^3$ (Casimir regime), the dark blue line is the universal quantum value of thermal conductance and the light blue line represents the best adjustment obtained with a power law in temperature of 1.8

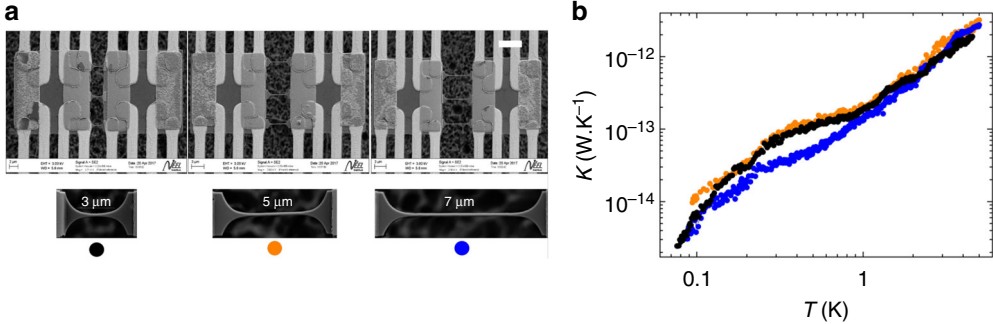

**Fig. 4** Thermal conductance measurements of nanowires having three different lengths as a function of temperature. **a** SEM images of the three different nanowires suspended between the membranes having a length between the two contacts of 3 μm (black dots), 5 μm (orange dots), and 7 μm (blue dots) for a constriction size $w$ = 2 μm. Scale bar 5 μm. **b** Thermal conductances of the three nanowires as a function of temperature. The measurements are mostly superposed showing that the phonon mean free path is at least >7 μm whatever the temperature

the temperature is lowered, the difference becomes larger than a factor of two; this could be attributed to intrinsic difference between the two nanowires as, for instance, the shape of the contact leading to different transmission coefficients between the one nanowire and the two nanowire samples. In any case, this observation remains relatively unexplained and requires more experimental investigations that go far beyond the original scope of this paper.

In order to distinguish the different regimes of phonon heat transfer, it is essential to compare the dependence in temperature of the thermal conductance to theoretical models. Indeed, $K(T)$ generally reveals the various mechanisms of phonon transport at low temperature. This is illustrated in Fig. 3b, where the measurements are compared to the different expected regimes of heat conduction at low temperature: boundary-limited heat transport (the Casimir regime) characterized by a cubic behavior in temperature (black line), and the quantum limit of fully ballistic transport characterized by a linear regime (dark blue line) only when the transmission coefficients are equal to one; the light blue line is a simple quadratic fit likely to describe thermal transport in amorphous materials like SiN when phonons are predominantly scattered by two level systems (TLS)[34–38]; in amorphous materials atoms or group of atoms may tunnel between two different equilibrium positions creating what is called a TLS. As it is shown in Fig. 3b, the dependency of thermal conductance on temperature is indeed mostly quadratic; a variation in temperature departing strongly from the universal value of thermal conductance[6–10,20].

Before going further in the analysis, it is crucial to investigate the true regime of phonon exchange between the two membranes. To probe this regime, and particularly if the thermal transport is ballistic, we performed thermal conductance measurements as a function of the length of nanowires.

This experimentally probes if the phonon mean free path is longer than the nanowire length. The different geometries are presented in Fig. 4a; the samples are constituted by three nanowires connecting the two membranes; nanowires with lengths ranging from 3 to 7 μm have been measured. As shown in Fig. 4b, the thermal conductances of these nanowires are largely superposed, showing no major difference in phonon heat transport between them. This is demonstrating that the phonon MFP is much larger than the effective length of the nanowires at all temperatures in the experiment. This means that phonons travel without any collision through the phonon waveguide. This demonstrates that phonon transport is ballistic at low temperatures. The absolute value of the thermal conductance is smaller than the one shown in Fig. 3a. The smaller constriction size of the contact to the membranes in the samples of Fig. 4, $a$ = 2 μm instead of 2.7 μm in Fig. 3, explains the reduced transmission; this trend is indeed expected from the Rego and Kirczenow[9] or Tanaka et al.[39] models and corroborated by our own calculations (see the numerical simulations in the Supplementary Fig. 1). Since the nanowire can be considered as a 1D phonon waveguide when $\lambda_{\mathrm{dom}}$ is much larger than the nanowire diameter $d$ (see Eq. (3)), this implies that, at least below a few degree kelvin, heat transport is in the right limit (ballistic and 1D) to observe quantum effects.

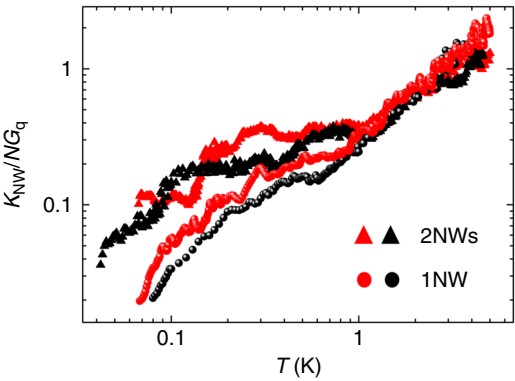

**Fig. 5** Normalized thermal conductance of nanowires. The thermal conductance of the nanowires $K_{NW}$ is normalized to $N$ times the quantum of thermal conductance $G_q$, $N$ being the number of channels. This value is obtained from the thermal conductance of the one and two nanowires shown in Fig. 3a divided by the universal quantum of thermal conductance times four channels for 1 NW and eight channels for 2 NWs. As in Fig. 3, red (black) markers are data collected heating from right to left (left to right). At low enough temperature (below a few kelvin), $K_{NW}/NG_q$ can be considered as equal to the Landauer transmission coefficient $\mathcal{T}$ as calculated in Eqs. (1) and (2)

It is however clear from Figs. 3b and 4b that the thermal conductance departs significantly from the expected universal value $G_q$. This is striking since we demonstrated above that transport is ballistic in a 1D phonon waveguide. This observation can be ascribed to transmission coefficients that are far from being close to unity. Consequently it is proposed to interpret these data in the light of the Landauer formalism. The transmission coefficients of the nanowire can be calculated from the measured thermal conductance presented in Fig. 3a, using Eq. (2) since $T_1 - T_2 < T$. In Fig. 5, the thermal conductance of the nanowires is normalized to $K_{NW}/NG_q$, $N$ being the number of channels ($N = 4$ for one nanowire and $N = 8$ for two nanowires); at low enough temperature $K_{NW}/NG_q$ is similar to the transmission coefficient $\mathcal{T}$ of Eq. (2). The transmission coefficients keep decreasing as the temperature decreases, showing that the thermal conductance does not reach the expected universal value, in a clear illustration of the breakdown of thermal conductance quantization.

## Discussion

The low transmission coefficients suggest that different mechanisms are at stake preventing a perfect transmission of phonons between the reservoirs. The intrinsic quantum value of thermal conductance of the 1D nanowire has to be seen as in series with the thermal resistances of the contacts. Above 4 K, $K_{NW}/NG_q$ is bigger than one, because more than the lowest four phonon modes are excited, leading to a thermal conductance bigger than the universal value. The transmission coefficients are identical for both 1 NW and 2 NWs from 5 to 1 K, and starts to differ at lower temperature. This could be the signature of a transition from 3D to 1D physics below 1 K; the dissimilarity may be attributed to different couplings and hence different thermal resistances between the nanowire(s) and the heat bath in the quantum limit.

The observed thermal resistance between the nanowires and the membrane reservoirs may have different origins. It could come either from the non-perfect coupling between the 1D nanowire and the 2D reservoirs[40], or from specific phonon scattering arising in the reservoirs. Calculations of the expected transmission coefficient in the catenoidal shape for the four

different phonon polarizations have been done by Tanaka et al.[39], in particular for SiN. The 1D–2D coupling could indeed have marked consequences on the phonon transport as demonstrated by Chalopin et al.[40], where numerical simulations clearly show that thermal transport is dominated by the contact resistances arising from a smaller number of excited modes in the membrane reservoir than in the nanowire. At low temperature, a Kapitza thermal resistance is expected to reduce the transmission coefficient below one. However, this is not the only possible origin for this thermal resistance. It has been shown recently that, in amorphous nanostructures, the phonon-TLS scattering is the limiting mechanism for phonon transport in the diffusive regime[38]. As shown in Fig. 3b, the overall temperature variation of the thermal conductance approaches $T^{1.8}$, a nearly quadratic dependence in good agreement with the expected thermal properties in amorphous structures at low temperature. It is commonly accepted that this characteristic behavior can be ascribed to the scattering of phonons on TLS. This scattering mechanism, arising in the junction or in the reservoir, is another possible explanation for the very low transmission coefficients between the ballistic nanowire and the membrane.

To summarize, these highly sensitive thermal measurements allow us to probe the thermal conductance of 1D phonon waveguides down to very low temperature. At these temperatures, it is shown that phonon exchange is ballistic between the two reservoirs. In this quantum limit, the experimental data can be understood in the framework of the Landauer model through the calculation of the transmission coefficient. Even in the ballistic regime, thermal transport is affected by a reduced transmission coefficient far below the unity. Kapitza like thermal resistance[39,40], boundary conditions[10] or phonon scattering[41] in the reservoirs creates a breakdown of the thermal conductance quantization. This new set of data opens unsolved questions such as the exact origin of the low transmission and thermal contact resistance, along with the actual mechanisms of phonon scattering in the junction to the reservoir.

These findings severely impede the observation of the regime of maximum exchange of heat or information (the quantum of thermal conductance) in a one dimensional phonon waveguide as proposed by Pendry[6]. Whilst we expected to observe a linear thermal conductance in temperature close to the quantum universal value, a quadratic temperature dependence is dominant at temperatures below a few kelvin. This low exchange of heat even in the low temperature regime may have major consequences in the thermalization of many nanophononic or nanoelectronic systems at low temperatures. Finally, the tens-of-attowatt sensitivity achieved by this measurement platform opens up tremendous possibilities for the next generation of low temperature thermal measurements at the nanoscale. Maximizing transmission coefficients will be the next experimental challenge if one wants to address exciting issues like phonon thermal rectification or phonon coherent effects.

## Methods

**Sample fabrication**. The samples are microfabricated from 100 mm Si wafer having a 100 nm thick thermally grown stoichiometric silicon nitride layer on top. The electrical superconducting leads that connect the transducers (heaters and thermometers) are in niobium titanium (100 nm) and gold (20 nm), they are made by laser lithography and deposited by sputtering. This NbTi/Au layer ensures very good electrical contact between the electrical leads and the transducers (heater and thermometer). Superconducting materials are used to reduce significantly the thermal conductance of the suspending beams far below the critical temperature $T_c$; $T_c$ is in most of the case above 7 K. This limits the thermal measurements below that temperature. After a laser lithography step, the copper heaters are deposited by e-gun evaporation and the niobium nitride (NbN) thermometers by sputtering using argon nitrogen plasma. All transducers are connected in a four-wire arrangement.

The nanowires linking the two membranes are nanofabricated using e-beam lithography. This allows a better definition of the shape and the boundary of the nanowires than laser lithography. Nanowires have a diameter in the center smaller than 100 nm; the catenoidal shape has been designed using the formula $A(x) = A_0\cosh^2(x/\lambda)$, where $A_0 = 10^{-7}$ m is the size of the thinnest part of the nanowire and $\lambda = 1.2$ μm is the characteristic length of the catenoide. An aluminum mask is deposited by e-gun evaporation at low temperature (100 K) to reduce the size of the grains. The silicon nitride layer is then structured by $SF_6$ reactive etching using the aluminum and photoresist as a mask; this step will determine the final geometry of the membrane, the nanowire and the suspending beams. After removing the aluminum layer by chemical etching, the whole measurement platform composed of the membrane, the nanowires and suspending beams are released by a gaseous chemical etching of the underlying silicon using an $XeF_2$ etching process. The full platform is consequently mechanically suspended and partly thermally isolated from the surrounding thermal bath. The chip is fixed on a copper-based printed circuit board (PCB) regulated in temperature. This experimental set-up is installed under vacuum in the measurement chamber.

**Experimental set-up.** The measurements are performed down to 50 mK in a dilution refrigerator; the PCB can be regulated in temperature up to 10 K. All the electrical connections are equipped with a low pass frequency filter ($f_c = 50$ kHz). The current source and the preamplifier are very low drift (below 5 ppm °C$^{-1}$) and very low noise (below 1 nV $\sqrt{\text{Hz}}^{-1}$) at frequency lower than 1 kHz. The heaters have a resistance of ~50 Ω and the thermometers of few kilo Ω. The measurement of the thermal conductance is performed by heating up one membrane using the Joule effect of one heater: $\dot{Q}_h = R_h I_h^2$. The heating current $I_h$ is comprised between 10 nA and 10 μA depending on the temperature of the regulated stage of the PCB. During the measurements, the temperature of both membranes is continuously monitored with the NbN thermometers measured using an AC current of 5 nA at 27 Hz. The power dissipated in the thermometers is much smaller than $\dot{Q}_h$. Regarding the performance of the measurements, the smallest variation in temperature that the thermometers can capture is $\delta T \simeq 50$ K giving an absolute error below the attoWatt in power (see below the section on sensitivity for more details).

When one membrane is heated up, the temperature of the second membrane raised due to the circulating heat flow through the nanowires. The stabilization of the temperature of both membranes is obtained when the powers are balanced between the heat that flows through the nanowire and through the suspending beams. If $T_0$, $T_1$, and $T_2$ are the temperatures of, respectively, heat bath, membrane 1 and membrane 2, then we can define the two flows of heat from each membrane to the heat bath:

$$\dot{Q}_1 = K_{b1}(T_1 - T_0) \tag{5}$$

$$\dot{Q}_2 = K_{b2}(T_2 - T_0), \tag{6}$$

$K_{bi}$ being, respectively, the thermal conductance between membrane $i$ and heat bath. By knowing the exact temperature of each membrane along with the power dissipated, we have access to the full balance of the transported flow of energy through the nanowires and membranes, and hence to the thermal conductance of the nanowires. By changing which platform is heated, one can reverse the heat flowing through the nanowires and then probe the symmetry of heat transport in the nanowires.

This measurement method based on two membranes having different temperatures permits accessing the thermal properties of a 1D phonon waveguide in the ballistic limit. In this limit, local temperature along the nanowire is not any more a relevant physical concept. At these very low temperatures (100 mK) competing methods like the $3\omega$ cannot be used because the temperature has to be known all along the suspended nanostructures[42]. Here, the temperature will be well defined only on the membrane considered as a sufficiently large thermodynamic reservoir.

**Thermal model of the measurement platform.** Here we derive the basic equations ruling the heat balance of the platform constituted by the two membranes, the nanowires and the suspending beams. This will permit the determination of the thermal conductance of the nanowires. We can write down the thermal balance by starting from the heating power dissipated by the heater:

$$\dot{Q}_h = R_h I_h^2 = \dot{Q}_1 + \dot{Q}_{NW} \tag{7}$$

where $\dot{Q}_{NW}$ is the heat flow through the nanowires. Similarly, this heat flow is related to the thermal conductance of the nanowires through:

$$\dot{Q}_{NW} = K_{NW}(T_1 - T_2), \tag{8}$$

in which $K_{NW}$ is the thermal conductance of the nanowires. In the steady state configuration, the system can be considered as in a stationary regime so the heat flux between the membrane and the heat flowing from the second membrane to the heat bath can be equaled: $\dot{Q}_{NW} = \dot{Q}_2$. Using this equation, the thermal

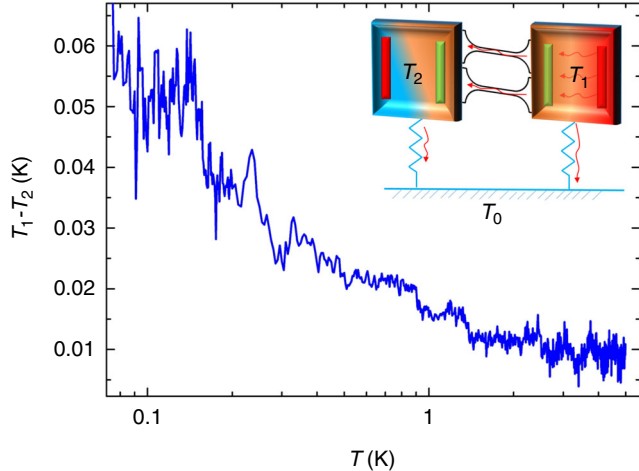

**Fig. 6** Typical temperature gradient measured between the two membranes during an experiment of thermal conductance as a function of temperature. This temperature gradient is smaller than 30 mK above 200 mK. In inset, schematic of experimental set-up

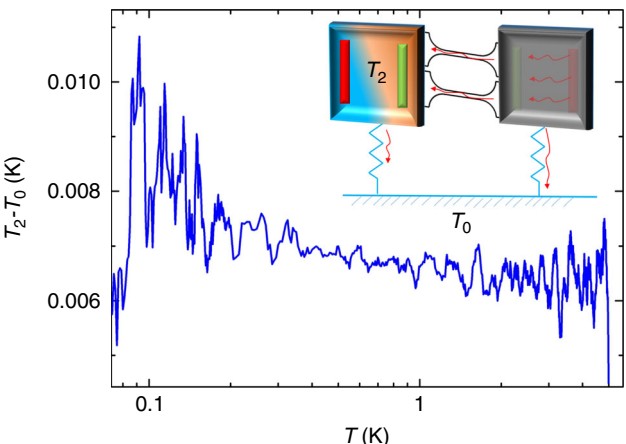

**Fig. 7** Typical temperature gradient measured between the second membrane and the thermal bath (baseline) during an experiment of thermal conductance as a function of temperature. The temperature gradient is smaller than 10 mK. In inset, schematic of measurement configuration

conductance of the nanowires can be expressed by:

$$K_{NW} = K_{b2}\left(\frac{T_2 - T_0}{T_1 - T_2}\right) \tag{9}$$

if the conditions $T_1 - T_2 \ll T_0$ and $T_2$ close to $T_0$ are fulfilled. By monitoring $T_0$, $T_1$, and $T_2$ (only $T_0$ is regulated) and having calibrated $K_{b2}$ as a function of temperature, $K_{NW}$ will be obtained as a function of $T_0$. $K_{b2}$ is measured on a similar sensor without the presence of nanowires (see Fig. 2d). The measurement of $T_1 - T_2$, $T_2 - T_0$ and the calibration of $K_{b2}$ are presented in the following figures: Figs. 6–8.

**Signal-to-noise ratio and attoWatt sensitivity.** One can estimate the expected signal to noise ratio of the thermal conductance measurement by using the law of propagation of uncertainty from Eq. (9):

$$\left(\frac{\delta K_{NW}}{K_{NW}}\right)_{\text{Theor}} = \sqrt{\left(\frac{\delta T_2}{T_2}\right)^2 + \left(\frac{\delta T_0}{T_0}\right)^2 + 2\left(\frac{\delta(T_1 - T_2)}{T_1 - T_2}\right)^2} \tag{10}$$

There is no added noise from $K_b$ since the input value comes from the fit of the measurement shown in Fig. 8. If the only limit is the Johnson–Nyquist noise of the thermometer, the theoretical error in temperature measurements is given by $\left(\frac{\delta T}{T}\right) = \left(\frac{\delta V}{V}\right)$. The Johnson voltage noise is calculated using the Nyquist relation $\delta V = \sqrt{4k_B T R_{Th}}$ in Volt.$\sqrt{\text{Hz}}^{-1}$, where $R_{Th}$ is the resistance of the thermometer.

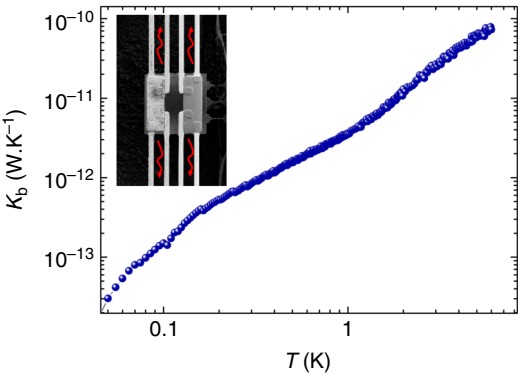

**Fig. 8** Measurement of the thermal conductance of the suspending beams $K_{b2}$ of one isolated membrane. In inset, the schematic of the measurement configuration

Then from Eq. (10), one can estimate the theoretical uncertainty on the thermal conductance: $\left(\frac{\delta K_{NW}}{K_{NW}}\right)_{Theor} = 4 \times 10^{-3} \sqrt{Hz}^{-1}$; the noise corresponding to a 4 kOhm thermometer resistance at 0.1 K is $\delta V = \sqrt{4 k_B T R_{Th}} = 0.15$ Volt $\sqrt{Hz}^{-1}$ for a voltage of $V = 40$ nVolt.

This theoretical value has to be compared to the experimental one extracted from the noise measurement on the thermal conductance: $\left(\frac{\delta K_{NW}}{K_{NW}}\right)_{Exp} = 1.5 \times 10^{-2}$ $\sqrt{Hz}^{-1}$, knowing that $K_{NW} = 10^{-14}$ W K$^{-1}$ at 100 mK, giving a noise on the thermal conductance of the order of $\delta K_{NW} \approx 10^{-16}$ W K$^{-1}$ $\sqrt{Hz}^{-1}$. This shows that the experimental error is only three times bigger than the expected theoretical one, attesting the low noise experiment. This also indicates the presence of other origins of noise like the preamp noise (1 nVolt $\sqrt{Hz}^{-1}$), the thermal drift of the equipments, etc. In any case, this helps to give a good estimate of the overall sensitivity of the experiment in terms of measurable power: $\delta P = \delta K_{NW} (T_2 - T_0)$, since $(T_2 - T_0)$ is of the order of 10 mK, the sensitivity in power is on the order of $15 \times 10^{-18}$ Watt $\sqrt{Hz}^{-1}$.

The phonon noise or also called the thermal fluctuation noise needs to be discussed. It is linked to the existence of thermal conductance connecting two heat baths. It can be described through a noise equivalent power (written down NEP) exchanged between these baths. The mathematical formula giving the phonon noise is[43]:

$$\text{NEP}_{ph,j,bi} = \sqrt{4 k_B T^2 K_{j,bi}} \tag{11}$$

where $K_{j,bi}$ represents the thermal conductance of either nanowires ($j$) or the suspending beams (bi). Therefore, in our measurement platform, two sources of thermal fluctuation noise exists coming from the exchange of phonons between the membranes and the heat sink, and between the membranes themselves. The numerical calculation using the measured values of thermal conductances gives $\text{NEP}_{phonon-j} = 9.18 \times 10^{-20}$ W $\sqrt{Hz}^{-1}$, and $\text{NEP}_{phonon-bi} = 2.73 \times 10^{-19}$ W $\sqrt{Hz}^{-1}$. The total $\text{NEP}_{ph}$ is equal to:

$$\text{NEP}_{ph,total} = \sqrt{\sum_j \text{NEP}_{ph,j,bi}^2} \tag{12}$$

$$= \sqrt{\text{NEP}_{ph,j}^2 + \text{NEP}_{ph,bi}^2} \tag{13}$$

The numerical computations give $\text{NEP}_{ph,total} = 0.3 \times 10^{-18}$ W $\sqrt{Hz}^{-1}$; a power smaller than the power sensitivity of the experiment (as shown above). Consequently, the temperature fluctuations created by the exchange of phonons are much smaller than what can be measured in that experiment, then the impact of the phonon noise on the results of the present work can be neglected.

**Negligible heat loss by radiation**. Experiment at low temperature has a significant difference with room temperature one: the contribution of radiation to the thermal balance becomes progressively negligible as the temperature is lowered. In order to show this, we can estimate the thermal power exchanged between nanowire and calorimeter walls:

$$P = \varepsilon \sigma S (T_1^4 - T_0^4) \tag{14}$$

where $\varepsilon$ is the emissivity (taken equal to 1 here), $\sigma$ is the Stefan-Boltzmann constant and $S$ the surface of exchange between the membrane at temperature $T_1$

and the calorimeter wall at temperature $T_0$. Using Eq. (14), $P$ is estimated to be on the order of $10^{-16}$ Watt for a temperature gradient of 4 K, a very small power as compared to the thermal conductance to the heat bath. However, for more secure measurements, a screen has been installed between the sample and the calorimeter walls thermally coupled to the temperature of the baseline. This is reducing significantly the temperature gradient resulting in a power exchange through radiation smaller than $10^{-21}$ Watt. As a consequence, any parasitic thermal path through radiation heat loss can be excluded in this experiment.

## Data availability

The data that support the findings of this study are available from the corresponding author on reasonable request.

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

## Acknowledgements

We thank the micro and nanofabrication facilities of Institut Néel CNRS: the Pôle Capteurs Thermométriques et Calorimétrie (E. André, G. Moiroux, P. Lachkar and J.-L. Garden) and Nanofab (L. Abbassi, S. Dufresnes, B. Fernandez, T. Fournier, G. Julié, and J.-F. Motte) for their help in the preparation of the samples. We have also benefited from the support of the Pole Cryogenie and Pole Electronique. O.B. and E.C. acknowledges the financial support from the ANR project QNM Grant No. 040401, the European projects MicroKelvin EUFRP7 Grant No. 228464 and MERGING Grant No. 309150.

## Author contributions

O.B. conceived the project with the help of E.C.; A.T., K.L., T.C., and O.B. designed and did the micro and nanofabrication of the sensors. A.T., E.C., and O.B. performed the low temperature thermal conductance measurements. N.M. performed the numerical simulations based on Green's functions. A.T., K.L., E.C., N.M., and O.B. interpreted the experimental and simulation results. A.T., E.C., and O.B. wrote the manuscript with the help of N.M.

## Additional information

**Competing interests:** The authors declare no competing interests.

