## [Peer Review File · Nature Communications]

Editorial Note: This manuscript has been previously reviewed at another journal that is not operating a transparent peer review scheme. This document only contains reviewer comments and rebuttal letters for versions considered at Nature Communications. Mentions of prior referee reports have been redacted.

Reviewers' comments:

Reviewer #2 (Remarks to the Author):

I find the measurements and their sensitivity interesting enough so that publication in Nature Communications might be feasible, provided the authors first respond to my comments from the previous review that I repeat below, together with some additional comments on the noise analysis, which has been presented very loosely and would have to be corrected prior to publishing of this work anywhere.

My comments from the previous round are:

[Redacted]

The comments on the Methods section:

a) (Page 13) The authors mention that the "smallest variation in temperature that the thermometers can capture is $\Delta T \sim 50 \mu\text{K}$ giving an absolute error below the attoWatt in power". Unless the authors' setup has $1/f^2$ noise, increasing the averaging time should improve the sensitivity to temperature changes, so this statement is meaningless.

b) (Page 13, paragraph above Eq. (5)). The authors mention "circulating" heat flow. Why is it circulating? On average, it just flows from hot to cold (2nd law of thermodynamics).

c) (Turn of page 13/14) I disagree with the statement that the 3w method requires the knowledge of the temperature all along the suspended nanostructure. Why would this be the case? As in the heat balance analysis, you could take the suspended nanostructure as a black box with some heat conductance (to be determined), and apply the analysis as usual. The only possible problem would arise outside the linear regime where the temperature variation would be large compared to the absolute temperature, but then also the thermal conductance might be ill-defined.

d) (Pages 14-16) In this discussion, the main factor to the sensitivity of the measurement to changes in the heat conductance comes from the voltage noise across the thermometer contact, giving an estimated relative sensitivity of the order of the voltage noise divided by the average measured voltage. The zero-frequency voltage noise (square root of the power spectral density) has units of $\text{volts}/\sqrt{\text{Hz}}$, which means that also the relative sensitivity has to have units of $1/\sqrt{\text{Hz}}$. In practice this means that increasing the measurement time by a factor of 100 improves the sensitivity by a factor of 10. Therefore, throughout the paper the sensitivity should be given in these units, not by a dimensionless number!

Please note that the equation for ΔV has an error of a factor of 10^9 on page 15. It should be in $\text{nV}/\sqrt{\text{Hz}}$, not in $\text{V}/\sqrt{\text{Hz}}$. When I calculated it, I got $0.15 \text{ nV}/\sqrt{\text{Hz}}$, not 0.16 (the more precise number is 0.1486).

As a small correction to the averaging time issue, the authors have added the sentence "This sensitivity corresponds to an averaging over one minute". This is a bit confusing. With the estimate that they do in the text, they should get to $1 \text{ aW}/\sqrt{\text{Hz}}$, i.e., they should get a sensitivity of aW already by averaging over one second. Of course, there may be $1/f$ noise limiting the actual sensitivity in a different way than zero-frequency noise, but then that should be included in the discussion.

[Redacted]

We would like first to thank the referees for their second round of reports. It was indeed helpful to further improve the presentation and the clarity of our paper. We think that we waved any possible confusions with previous works.

We hope that now our manuscript is reaching the high level required for publication in the Nature Communications journal regarding the originality, and the significance of our new set of experimental results.

Please find below the answers we can provide for each comment and/or question along with a list of changes that have been made in the new version to take into account the referees' remarks.

Reviewer #2

We thank the referee #2 for his valuable comments. First, we indeed should have cited and commented the work done by Koppinen *et al.* Second, clarification on what is “quantum” in our experiment was also necessary. The Methods has been also improved thanks to the comments on the noise and sensitivity. We have followed the recommendations of the referee and we now present the performance of our sensor per $\sqrt{\text{Hz}}$.

[Redacted]

The authors: we agree with the referee that, indeed, it is perfectly appropriate to cite and comment the nice work done by Koppinen and Maasilta ten years ago. They were indeed measuring the thermal transport in the same limit as in this work. However, as we will show below and like with ref 9 (Schwab et al), this experiment contains intrinsic significant differences from the present work that fully justify the publication of our paper.

Even, if they indeed see ballistic signature of heat transport, there is two major differences from the Koppinen's experiment and ours:

1-in the present work, the phonon waveguide is not covered by a metallic coating that may perturb the phonon transport. In the ref 9, the nanowire is covered by niobium and in the Koppinen's experiment by copper (a normal metal); this extra-coating will definitely serve as parasitic thermal path, this is not the case in our experiment.

2-in the Koppinen's work, as mentioned by the authors themselves, the contact between the phonon conductor and the heat bath is abrupt reducing *de facto* the transmission coefficient. Any comparison of our work to Koppinen *et al.* is then difficult (very different contact geometry) and, consequently, comment on the quantization (or not) of the thermal transport is rather problematic in their case.

As required by referee #2, we list below again the major new aspects of our experiment as compared to previous studies:

1-An innovative experimental design giving improved quality of the data set in this regime of low temperature heat conduction. Better confidence can be attributed to the thermal measurements and favor new theoretical modelling and discussions.

2-We show a breakdown of the quantization of thermal transport, unexpected in this limit (1D +ballistic): optimization of the geometrical profile following Rego *et al.* and Tanaka *et al.* recommendations, adequate temperature range, measurements done on 1D phonon waveguides. The transmission coefficients are much smaller than unity at low temperature (below 1K).

3-We show that, since the transport is ballistic, the thermal resistance is coming from the junction and the reservoir. This thermal resistance dominates the phonon transport hiding a potential quantization of the phonon transport in the 1D structure.

Modification of the text: we have corrected the text and made reference to the work done by Koppinen et al.

[Redacted]

The authors: concerning the point raised by the referee regarding quantization, we fully agree that our experimental data cannot demonstrate the quantum aspect of phonon transport but only ballistic transport in a 1D structure. More than that, we actually show a breakdown of the quantization. We however claim that we built the experiment in order to measure the thermal transport in the quantum limit. In other word, we have prepared the experiment to fulfil all the requirements to obtain quantum heat transport, even if the experimental results do not demonstrate quantization, which is the crucial point of the paper.

Modification of the text: we have modified the text accordingly to remove any confusion in the understanding. “Quantum regime” may be misunderstood, it has been removed in the abstract and replaced by ballistic 1D whenever it was possible, especially in the discussion of the results page 9.

[Redacted]

The referee is questioning the appropriateness of some of our prospective given in the abstract. It is indeed perfectly right that implications of our measurements will not impact all types of nanoelectronic systems. Our point was to raise awareness on the implications of the results of this work on innovative thermal management including, potentially, phonon based exchange of information. The goal number iii is indeed satisfied by this work since contact resistance is a very important subject for potential applications, even if more experiments are needed to really control this aspect.

Modification of the text: to avoid any reference to far-fetched consequences of this work, we changed the text of the abstract to focus on heat management or nanophononic devices only.

We address below the various answers to referee #2 questions regarding the Methods.

Question (a) and (d): a) (Page 13) The authors mention that the "smallest variation in temperature that the thermometers can capture is $\Delta T \sim 50 \mu\text{K}$ giving an absolute error below the attoWatt in power". Unless the authors' setup has $1/f^2$ noise, increasing the averaging time should improve the sensitivity to temperature changes, so this statement is meaningless.

d) (Pages 14-16) In this discussion, the main factor to the sensitivity of the measurement to changes in the heat conductance comes from the voltage noise across the thermometer contact, giving an estimated relative sensitivity of the order of the voltage noise divided by the average measured voltage. The zero-frequency voltage noise (square root of the power spectral density) has units of volts/sqrt(Hz), which means that also the relative sensitivity has to have units of 1/sqrt(Hz). In practice this means that increasing the measurement time by a factor of 100 improves the sensitivity by a factor of 10. Therefore, throughout the paper the sensitivity should be given in these units, not by a dimensionless number!

Author's answer: we followed the referee's recommendations and change the all the Methods to present the noise of the thermal conductance per Sqrt(Hz).

Question (b): (Page 13, paragraph above Eq. (5)). The authors mention "circulating" heat flow. Why is it circulating? On average, it just flows from hot to cold (2nd law of thermodynamics).

Author's answer: we agree with the referee that "circulating" is inappropriate, this term has been changed for "heat flows".

Question (c): c) (Turn of page 13/14) I disagree with the statement that the 3w method requires the knowledge of the temperature all along the suspended nanostructure. Why would this be the case? As in the heat balance analysis, you could take the suspended nanostructure as a black box with some heat conductance (to be determined), and apply the analysis as usual. The only possible problem would arise outside the linear regime where the temperature variation would be large compared to the absolute temperature, but then also the thermal conductance might be ill-defined.

Author's answer: we agree that our statement was not fully explained. What we meant was that the 3 omega method is not the most appropriated one for studying ballistic heat flow in nanowires. We have corrected the text to fit with this requirement, and moderate our judgment.

However, we are convinced that there will be significant intrinsic problems linked to the thermalization of the thermometer on top and the nanowire (Kapitza resistance), along with the parasitic thermal path of the thermometer that will seriously impede the use of the 3 omega method at very low temperature for thermal conductance measurement. Moreover, the thermometer might be at a very different temperature than the nanowire itself. The problem of the 3 omega at very low temperature does reside only in establishing the proper adequate mathematical model to extract the thermal transport, there is also thermal constraints that will be difficult to overcome.

Reviewer #4

We thank the referee for her/his thorough reading of our paper. We have follow several of her/his recommendations that indeed improve the quality of the manuscript. However, we do not fully share her/his opinion on the necessity of performing extensive modelling, especially phenomenological model (TLS) will not carry much improvement of the present understanding. We think that new elaborated theoretical models are indeed needed, but this goes far beyond the scope of our work. We have the feeling that the present models (Rego *et al.* and Tanaka *et al.*) do not grab the full physics. They could be too optimistic in evaluating the value of the contact resistance.

[Redacted]

The authors: we agree that, for the time being, our work is more qualitative than yet quantitative, but this does not wave its broad interest for the physics reader. It will even motivate further theoretical works.

However, we have taken the recommendations of the referee very seriously. A modelling of the effect of the size of the constriction has been made using Green's function method that explains qualitatively the difference of thermal transport between figure 3 and figure 4. Additional information can now be found in the Supplemental Information. We have also modified figure 5 to wave any possible confusion regarding transmission coefficient versus K_{NW}/NG_q where N is the number of channels of heat conduction.

[Redacted]

The authors: the referee is indeed correct. Our measurements depart significantly from the one obtained by Schwab et al (now ref 9). This is the major point of our paper, we show that no quantization at all is observed in our numerous sets of data despite ballistic transport in a 1D system; this is the major disagreement with Schwab's results.

As explained in our previous rebuttal letter, measurements from ref 9 above 0.3 K should be probably disregarded. Since the temperature range over which the data are presented is much reduced, it is difficult to attribute a full credit to measurement below 0.3 K; we then think that the problem is really open.

As mentioned in the text, we attribute this low value of transmission to very bad coupling between the nanowire and the membrane (heat sink) that still needs a full theoretical treatment, even if we know from Tanaka *et al.* paper that for some configurations the transmission coefficient can be much smaller than 1 (see for instance figure 5 of Tanaka for $\lambda=1\mu\text{m}$).

[Redacted]

The authors: we choose to show the 2 nanowires and 1 nanowire sets of data for the sake of completeness even if we knew that indeed a more complex physics may occur that just simple transmission through one nanowire. We do not have an explanation yet for this discrepancy; this is now clearly mentioned in the new version. However, we would like to highlight that this result is not the main message of the paper.

[Redacted]

The authors: this is indeed correct. The transmission coefficient between the nanowires and the heat bath is lower in Fig. 4 than in Fig. 3. We attribute this difference to a lesser coupling between the nanowire and the heat sink. This is expected since the size of the contact is much smaller ($2\mu\text{m}$ instead of $2.7\mu\text{m}$) in the samples presented in Fig. 4. This is now supported by the numerical simulation done in collaboration with Natalio Mingo using Green's function method. From this rough modelling, we can illustrate the clear trend that for a given contact (membrane), a reduced size of constriction between the nanowire and the membrane will give a reduced phonon transmission, as observed in our experiment.

Modifications: the text has been consequently modified, and a Supplemental Information is added to present the numerical simulations separately.

[Redacted]

The authors: the TLS scenario is not contradictory with the ballistic transport since we only claim that the scattering occurs in the junction or in the membrane and not in the nanowire. This is justified since the membrane is a much larger reservoir than the nanowire, the number of TLS is by definition larger and then may scatter phonon with a much higher probability. We proof read again the paper and we do not see where this misunderstanding may have occurred.

[Redacted]

The authors: we agree with the referee that plotting the transmission coefficient instead of the normalization versus N quantum of conductance is confusing, even if it gives the same information at low temperature.

Modifications: we have corrected the figure 5 and now plot K/NGq removing any possible misinterpretation of the data.

Reviewer #5

[Redacted]

The authors: we thank the referee for her/his positive evaluation of our work. As mentioned in our answer to reviewer #2, the data clearly show ballistic phonon transport in a 1D nanowire, but indeed there is no experimental signature of quantum transport since we demonstrate a breakdown of the thermal conductance quantization. This is now clearly explained in the text, excessive reference to “quantum transport” has been changed for “ballistic 1D transport” which is indeed more accurate. We hope that now our point is clearer.

Reviewers' comments:

Reviewer #4 (Remarks to the Author):

Based on the authors' responses, I am not fully convinced that the authors quite understand the physics underlying their results very well. Also, from an impartial referee's point of view, it is not quite fair to say others (specifically Schwab et al, ref 9) are wrong because the authors' results are different from others' (within the temperature range of less than 0.3K), unless the authors can give convincing theoretical argument to show why their results are reasonable, in particular, why their observed transmission coefficient is much less than one. In fact, Schwab's results can be well understood by the present models (Rego et al. and Tanaka et al.). The authors speculated that "We have the feeling that the present models (Rego et al. and Tanaka et al.) do not grab the full physics. They could be too optimistic in evaluating the value of the contact resistance." However, that authors did not provide any specifics (i.e., what physics Rego et al and Tanaka et al were missing and how?).

I agree with referee 5 that the manuscript does provide a very good set of data that could be interesting for the broader audience. I would suggest one of the two ways for the authors to revise their paper to a level that is acceptable in my opinion.

1. provide more convincing theoretical argument to show why their results are reasonable, and why Rego et al's models are not adequate.

OR.

2. significantly tone down their language, and present the paper as providing a new set of data on an old measurement (with improved platform) and pose several open questions.

We thank the referee #4 for her/his last report showing that our experiment and results can be accepted for publication under certain modifications.

Please find below the answers to the referee's comments along with the list of changes that have been made in the new version to take into account the referee's remarks.

Reviewer #4

Based on the authors' responses, I am not fully convinced that the authors quite understand the physics underlying their results very well. Also, from an impartial referee's point of view, it is not quite fair to say others (specifically Schwab et al, ref 9) are wrong because the authors' results are different from others' (within the temperature range of less than 0.3K), unless the authors can give convincing theoretical argument to show why their results are reasonable, in particular, why their observed transmission coefficient is much less than one. In fact, Schwab's results can be well understood by the present models (Rego et al. and Tanaka et al.). The authors speculated that "We have the feeling that the present models (Rego et al. and Tanaka et al.) do not grab the full physics. They could be too optimistic in evaluating the value of the contact resistance." However, that authors did not provide any specifics (i.e., what physics Rego et al and Tanaka et al were missing and how?).

We have carefully proofread our manuscript to expunge the text from any frontal opposition to Schwab's et al. work (see comment #2). At the same time, we provided a 2.5 page supplementary material in pdf format, where we explained that:

1-Rego's (and Tanaka's) model can explain Schwab's experiment, and **it can also explain ours**, but 2-it is unclear that some assumptions in Rego's and Tanaka's models are a fair representation of the real experimental structure (see our more detailed response below).

Therefore, we are neither attacking Schwab's results, nor Rego's or Tanaka's models, but we are nevertheless providing physical arguments (in our supplemental information) to point out that a fair quantitative comparison between experiment and theory would require more complex modeling beyond that carried out by Rego or Tanaka.

I agree with referee 5 that the manuscript does provide a very good set of data that could be interesting for the broader audience. I would suggest one of the two ways for the authors to revise their paper to a level that is acceptable in my opinion.

1. provide more convincing theoretical argument to show why their results are reasonable, and why Rego et al's models are not adequate.

Regarding point #1: in the supplemental information (SI) from our previous submission we provided theoretical arguments explaining why our observed transmission coefficients are much less than one. We also backed those arguments by a calculation with a model equivalent to Rego and Kirczenow's. To summarize what we have explained in the SI: the transmission across the constriction can easily be affected by structural features that are rather far from the constriction's narrowest point. In particular, there is a noticeable reduction in transmission when the catenoidal constriction is abruptly plugged into a much thicker contact, leading to a step-like feature in the profile (see figure 1 of the supplementary information). Such an abrupt feature is present in our experimental structures.

In the SI we also explain that our model, and equivalently that of Rego & Kirczenow's, makes two simplifying assumptions: 1) it approximates the waves in the structure by a scalar field instead of using the true vector displacement field, and (2) these waves are considered to depend only on one spatial direction along the structure, whereas in reality they should be a function of the three spatial coordinates. The first simplification thus prevents mode inter-conversion at the constriction. The second simplification makes the model inadequate to deal with constrictions that are sandwiched between two extended surfaces perpendicularly to the axial direction. Rego and Kirczenow considered a structure with a constriction into an otherwise constant-thickness, infinitely extended wire. Approximation (2) is adequate for such an ideal wire-like structure, but experimentally one will always need to plug the wire to something else. When this is done, the waves will no longer vary only in one spatial direction: for example, if the wire is plugged to a large single crystal the waves will propagate radially away from the junction into the crystal. Such aspect cannot be dealt with using Rego & Kirczenow's model, but requires a full 3D treatment that is rather involved and beyond the scope of this article. Although Tanaka's model relaxed assumption 1 and discussed inter-mode conversion to some extent, the system's geometry was also an infinitely long wire with a constriction, which does not correspond to the actual case for the experimental contacts.

For these reasons, we added a word of caution in our SI, stressing the need for fully 3D simulations if a quantitative comparison is done between theory and experiment. The infinite wire geometries used in Rego&Kirczenow's and Tanaka's papers, and in our SI model, are not a fair representation of the real contacts present in our experiment. Such models indeed predict an influence of structural features like the second abrupt contact present in our structures, and we have shown that they qualitatively explain the transmission reduction that we observe experimentally. However, a quantitative comparison requires going beyond the approximations in those models.

OR.

2. significantly tone down their language, and present the paper as providing a new set of data on an old measurement (with improved platform) and pose several open questions.

Regarding point #2: We have made two modifications in the main text to tone down our claims and wave any frontal opposition to Schwab's previous results:

1-we removed one sentence in the abstract "This high resolution experimental achievement gives new insight of heat transport in the ballistic 1D regime, a scientific goal that has remained uncertain till now."

2-we added an important note in the summary of the findings of this paper, letting clearly the door open for questions:

"This new set of data opens unsolved questions like the true origin of the low transmission and thermal contact resistance, along with the actual mechanisms of phonon scattering in the junction to the reservoir; this will need to be addressed in future works"